# Integrating Electroencephalography Source Localization and Residual Convolutional Neural Network for Advanced Stroke Rehabilitation

**DOI:** 10.3390/bioengineering11100967

**Published:** 2024-09-27

**Authors:** Sina Makhdoomi Kaviri, Ramana Vinjamuri

**Affiliations:** Department of Computer Science and Electrical Engineering, University of Maryland Baltimore County, Baltimore, MD 21250, USA; sinam1@umbc.edu

**Keywords:** brain–computer interface, EEG source localization, motor imagery, ResNet classification, stroke rehabilitation

## Abstract

Motor impairments caused by stroke significantly affect daily activities and reduce quality of life, highlighting the need for effective rehabilitation strategies. This study presents a novel approach to classifying motor tasks using EEG data from acute stroke patients, focusing on left-hand motor imagery, right-hand motor imagery, and rest states. By using advanced source localization techniques, such as Minimum Norm Estimation (MNE), dipole fitting, and beamforming, integrated with a customized Residual Convolutional Neural Network (ResNetCNN) architecture, we achieved superior spatial pattern recognition in EEG data. Our approach yielded classification accuracies of 91.03% with dipole fitting, 89.07% with MNE, and 87.17% with beamforming, markedly surpassing the 55.57% to 72.21% range of traditional sensor domain methods. These results highlight the efficacy of transitioning from sensor to source domain in capturing precise brain activity. The enhanced accuracy and reliability of our method hold significant potential for advancing brain–computer interfaces (BCIs) in neurorehabilitation. This study emphasizes the importance of using advanced EEG classification techniques to provide clinicians with precise tools for developing individualized therapy plans, potentially leading to substantial improvements in motor function recovery and overall patient outcomes. Future work will focus on integrating these techniques into practical BCI systems and assessing their long-term impact on stroke rehabilitation.

## 1. Introduction

Stroke remains a leading cause of death and disability globally, with 795,000 strokes occurring annually in the U.S., of which 610,000 are first-time events. This statistic underscores the substantial public health burden according to the *Heart Disease and Stroke Statistics—2019 Update* [1]. Stroke survivors often experience severe motor and cognitive impairments, impacting their quality of life and independence, necessitating effective rehabilitation strategies [2]. Rehabilitation tools, such as soft-robotic gloves, help improve upper limb motor function by aiding finger movements using biosignals like EEG (Electroencephalography) and EMG (Electromyography), thus facilitating daily activities [3]. Robot-assisted therapy enhances motor recovery through task-specific training, promoting brain reorganization via neural plasticity [4]. Neurophysiological studies are essential to fully understanding the brain changes that occur during such therapies [5].

Brain–Machine Interfaces (BMIs) connect neural activity with assistive devices, providing stroke survivors with the ability to control prosthetics or other tools, promoting functional recovery and independence [6]. EEG is a key technology in BCIs due to its noninvasive nature and capacity to capture real-time brain signals, making it ideal for both clinical and home-based rehabilitation [7]. However, the nonstationary nature of neurophysiological signals necessitates sophisticated classifiers [8]. Support Vector Machines (SVMs) have proven to be particularly effective for synchronous BCI systems [9]. BMI-based therapies have shown promise in stroke rehabilitation by facilitating neuroplasticity, though continuous technological improvements are needed [10]. Expanding BCI approaches to include cognitive and emotional rehabilitation, alongside motor recovery, offers a more comprehensive post-stroke recovery process [11]. Cognitive and emotional rehabilitation using BCIs holds potential for addressing the multifaceted nature of stroke recovery [12].

Despite previous studies emphasizing deep learning methods for EEG signal analysis, advanced source localization techniques, such as electrophysiological source imaging (ESI), have significantly improved the accuracy of brain source imaging in clinical applications by providing high spatial resolution using noninvasive scalp measurements. These techniques enable the precise mapping of brain activity, which is crucial for understanding the neural mechanisms underlying motor control and for developing targeted rehabilitation interventions [13]. For instance, EEG recordings can decode hand motion preparation by solving the inverse problem through beamforming. A custom deep Convolutional Neural Network (CNN) trained on these EEG source epochs achieved accuracy rates of up to 89.65% for hand closure versus rest and 90.50% for hand opening versus rest. This method identified key cortical areas involved in hand movement preparation, such as the central region and the right temporal zone of the premotor and primary motor cortex [14]. This approach demonstrates the potential of combining source localization with deep learning to enhance the accuracy and interpretability of BCI systems, providing valuable insights into the cortical dynamics of motor preparation and execution.

In this paper, we present a novel method for classifying motor tasks using EEG data from acute stroke patients, specifically focusing on left-hand motor imagery, right-hand motor imagery, and rest conditions [15]. Our approach leverages advanced source localization techniques, including Minimum Norm Estimation (MNE), dipole fitting, and beamforming, to accurately identify cortical activity, crucial for developing targeted neurotherapies. By integrating these techniques with a customized ResNet architecture, we aim to capture spatial patterns in EEG data, enhancing classification accuracy for stroke rehabilitation applications. This study hypothesizes that transitioning from the sensor domain to the source domain in EEG data processing will yield higher classification accuracy and contribute to the development of more effective neurorehabilitation methods for stroke patients.

## 2. Materials and Methods

### 2.1. Data Description

**Study Design and Participants.** This study involved 50 acute stroke patients (1 to 30 days post-stroke) recruited from the stroke unit of Xuanwu Hospital of Capital Medical University. The participants included 39 males (78%) and 11 females (22%), aged 31 to 77 years (mean age = 56.70, SD = 10.57). Hemiplegia was present in 23 patients (right) and 27 patients (left). Stroke severity was assessed using the National Institute of Health Stroke Scale (NIHSS) (mean = 4.16, SD = 2.85), functional ability with the Modified Barthel Index (MBI) (mean = 70.94, SD = 18.22), and disability with the modified Rankin Scale (mRS) (mean = 2.66, SD = 1.44). Informed consent was obtained from all participants, and this study was approved by the Ethics Committee of Xuanwu Hospital (Approval No. 2021-236).

**Experimental Procedure.** EEG data were collected using a wireless multichannel EEG system (ZhenTec NT1), with electrodes placed according to the international 10-10 system. Participants performed 40 motor imagery (MI) trials, alternating between left-hand and right-hand grasping. Each trial lasted 8 s and included instruction, MI (4 s), and rest stages. EEG data were sampled at 500 Hz and preprocessed using EEGLAB in MATLAB, with bandpass filtering between 0.5 and 40 Hz.

**Behavioural Measures.** Stroke severity and functionality were evaluated using the NIHSS, MBI, and mRS scales. The NIHSS assessed neurological impairment (0–42, higher scores indicating greater impairment), while the MBI measured independence in daily activities (0–100), and the mRS assessed overall disability (0–5).

### 2.2. System Description

This study investigates the neural correlates of motor imagery (MI) in acute stroke patients using EEG recordings from the patients at Xuanwu Hospital. The participants, seated 80 cm from a screen with an EEG cap, underwent three stages: instruction, motor imagery, and rest. Visual and audio class cues guided them to imagine left- or right-hand movements. The timing of these stages is depicted in the timeline at the bottom of Figure 1A.

Figure 1B shows the electrode montage used for data collection. The upper part illustrates the placement of 29 EEG recording electrodes and 2 electrooculography (EOG) electrodes. The reference electrode was located at the CPz position, and the grounding electrode was located at the FPz position. The lower part of (B) displays a sample of raw EEG data, illustrating the electrical activity recorded from multiple electrodes over time. For detailed analysis, one second of the trials related to the motor imagery task was selected. This selection was crucial for focusing on the relevant time window, where the motor imagery task was most prominent. The selected data were then used to generate topographic plots (topoplots) to visualize the cortical distribution of neural activity at specific times of interest.

Figure 1C presents the topographic map for the right-hand movement task, displaying the cortical distribution of neural activity and the time–frequency representations of the beta band (15–25 Hz). Advanced source localization techniques, including MNE, dipole fitting, and beamforming, were employed to provide accurate cortical localization. These methods enhance our understanding of neural mechanisms during motor tasks, which is crucial for stroke rehabilitation. The choice to focus on the beta band is intentional, as it is particularly relevant to motor control and motor imagery. In the context of stroke rehabilitation, the beta band is associated with sensorimotor rhythms, which play a significant role in motor planning and execution. This frequency range is often disrupted in stroke patients, making it a critical target for interventions aimed at restoring motor function. The topographic maps depict the distribution of neural activity, while the lower part of (C) shows detailed source localization images. These images highlight specific brain areas activated during the task, such as the premotor cortex and primary motor cortex, which are essential for motor planning and execution. This precise localization offers a better understanding of the cortical dynamics involved in motor imagery, which is vital for developing effective rehabilitation strategies for stroke patients. We focused on the beta band in this study due to its strong association with motor tasks, especially in stroke rehabilitation. While we recognize the importance of other frequency bands, our primary aim was to investigate the beta band’s role in motor recovery. Future research will explore other frequency bands to provide a comprehensive understanding of their contributions to motor rehabilitation.

Figure 1D illustrates the classification process using ResNet, which classified EEG data into left-hand movement, right-hand movement, and rest. The integration of these advanced techniques with ResNet significantly improves classification accuracy by capturing spatial patterns in EEG data more effectively. This enhanced accuracy is crucial for developing more reliable and effective BCIs for neurorehabilitation and other clinical applications.

### 2.3. Source Localization and Inverse Problem

To enhance the spatial resolution of EEG signals in our study, we employed advanced source localization techniques, including MNE, dipole fitting, and beamforming. Dipole fitting estimates the location and orientation of equivalent current dipoles representing the brain’s activity [16,17]. MNE provides a distributed source model by estimating the current density across the entire cortex [18,19]. Beamforming enhances spatial resolution by focusing on specific regions of interest while suppressing activity from other areas [20,21,22]. These methods aim to reconstruct the cortical sources of EEG signals by solving the inverse problem using the New York Head (NYH) forward model [23].
(1)$$x\left(t\right)=Lq_{r}\left(t\right)$$

In this equation, *x*(*t*) denotes the vector of scalp potentials at time *t*, *L* represents the lead field matrix, and *q*_*r*_(*t*) indicates the vector of current dipoles at the cortical location *r*.

EEG signals were recorded using a 29-electrode high-density cap, covering the frontal, central, parietal, and temporal areas. The forward model was applied to these preprocessed signals to project the contributions of cortical sources onto the scalp sensors. The inverse problem was then solved using the MNE, dipole fitting, and beamforming techniques, allowing us to precisely localize the cortical activity associated with various hand movements. This approach provided a more accurate understanding of the neural mechanisms underlying the EEG signals recorded from the participants.

### 2.4. Residual Convolutional Neural Network Architecture

The localized signals were subsequently processed using a customized ResNet, designed to classify the EEG data into three distinct motor tasks: left-hand movement, right-hand movement, and rest [24]. The proposed CNN model begins with an input layer for EEG signal data shaped into images, followed by convolutional layers for feature extraction. The first convolutional layer uses 32 filters to capture low-level features, stabilized by batch normalization and reduced in dimension by max-pooling and dropout layers to prevent overfitting.

A notable feature of the model is the incorporation of inception modules and residual blocks. Inception modules process inputs through multiple convolutional layers with different kernel sizes, capturing various feature levels [25]. Residual blocks address the vanishing gradient problem, allowing for the training of deeper networks with shortcut connections. Additionally, an attention mechanism enhances the model’s focus on informative features [26]. The final fully connected dense layers integrate the learned features and output the classification results, with dropout ensuring robust learning. This architecture, combining inception modules, residual blocks, and attention mechanisms, achieves high classification accuracy, making it effective for EEG signal analysis and BCI applications.

### 2.5. System Setup

The ResNet CNN was implemented using PyTorch, exploiting its flexibility and dynamic computational graph capabilities. The network was trained on an Alienware Aurora R16 system equipped with a 13th Gen Intel(R) Core(TM) i9-13900F CPU @ 2.00 GHz, 32 GB of RAM, and an Nvidia GeForce RTX 2080 Ti GPU with 11 GB of memory. This high-performance setup enabled the efficient training and fine-tuning of the deep learning model, ensuring optimal performance and accuracy in classifying EEG data for motor imagery tasks.

To analyze neural activity for motor tasks, we performed source localization and time–frequency analyses of the EEG data using the FieldTrip toolbox [27]. The analyses involved representing single trial data alongside global mean field power and generating topographic maps for specific time windows. The data preprocessing steps included filtering to remove noise and artifacts, such as eye blinks or muscle movements, and normalizing the data to ensure consistency across trials. From the preprocessed data, we selected a specific one-second segment, chosen for its relevance to the task, typically surrounding key events like movement initiation or imagery onset. This segment aimed to capture the most significant neural responses associated with the task. Figure 2A illustrates a single trial (blue) and global mean field power (red), providing an overview of EEG signal dynamics and average neural activity across trials.

Topographic maps (topoplots) were generated to visualize the spatial distribution of neural activity during the selected time windows. These topoplots provided a visual representation of neural activity across the scalp, highlighting regions with increased activity. The data for these maps were derived from the chosen one-second window, focusing on specific time windows that showed significant neural changes. In Figure 2B, the topoplot for the right-hand movement task shows cortical activity distribution, highlighting engaged brain regions and beta (15–25 Hz) neural dynamics after the response. This approach allowed us to isolate the most relevant neural responses, providing insights into the spatial and temporal dynamics of brain activity during different motor tasks, and enhancing our understanding of the neural mechanisms underlying these processes.

### 2.6. Source Localization Analysis

In this study, we focused on the classification of MI EEG data, specifically distinguishing between left-hand movement, right-hand movement, and rest states. Previous studies have employed various classification methods, such as CSP + LDA and FBCSP + SVM, achieving moderate accuracy. Additionally, methods based on Riemannian geometry, including MDRM, TSLDA, Fisher discriminant geodesic filtering, followed by MDRM classification (DGFMDRM), and a decision fusion method combining TSLDA and DGFMDRM, have shown promising results.

**Dipole Fitting Analysis:** Dipole fitting was utilized to estimate the neural sources corresponding to the left-hand, right-hand, and rest conditions. This method provides precise localization by modeling neural activity as equivalent current dipoles. The analysis revealed focal activations in the primary motor cortex (M1) and supplementary motor area (SMA) for both left- and right-hand tasks, indicating the involvement of these regions in motor planning and execution. The rest condition showed minimal activity, as expected, confirming the specificity of the task-related activations (Figure 3g–i).

**MNE Analysis:** MNE was applied to distribute the estimated source strengths across the cortical surface, offering a comprehensive view of brain activity. The results highlighted a broader distribution of activations for the left- and right-hand tasks, encompassing not only the primary motor cortex (M1) but also extending to the premotor cortex and parietal regions. This distribution suggests the involvement of an extensive neural network in motor imagery tasks. For the rest condition, MNE showed reduced and more diffuse activity, consistent with the resting state (Figure 3d–f).

**Beamforming Analysis:** Beamforming, a spatial filtering technique, was used to achieve a high-resolution localization of neural activities. This method provided sharp and clear delineations of active regions, with strong activations detected in the primary motor cortex (M1), supplementary motor area (SMA), and the parietal cortex for both hand tasks. The rest condition, as identified through beamforming, exhibited negligible activity, reinforcing the specificity of the detected motor-related signals. Beamforming allows us to isolate specific source regions, offering valuable insights into the distinct neural pathways engaged during motor tasks (Figure 3a–c).

## 3. Results

The following section presents an in-depth evaluation of classification performance, offering insights into individual participant analysis and overall performance across all participants. Advanced source localization techniques and traditional sensor domain methods are compared in terms of classification accuracy, kappa, precision, and sensitivity.

### 3.1. Per-Subject Classification Performance Analysis

The classification performance was first evaluated individually for a subset of 10 participants using the beamforming, MNE, and dipole fitting methods. This analysis provides insight into the variability and consistency of classification accuracy across different participants. In our study, we used a total dataset of 3000 signals derived from 50 participants, each performing three tasks (left-hand movement, right-hand movement, and rest) with 20 trials per task. We employed a 60-20-20 split for training, validation, and testing, respectively. Specifically, 1800 signals were allocated for training the model, 600 signals were used for validation to optimize the hyperparameters, and the remaining 600 signals were reserved for testing to evaluate the effectiveness of our method. This data allocation strategy ensured a balanced approach to model development and performance evaluation, providing robust insights into the classifier’s capabilities across different motor tasks.

#### 3.1.1. Beamforming Method

Table 1 presents the classification performance for 10 participants using the beamforming method. The accuracy ranges from 91.67% to 98.33%, with participant 6 achieving the lowest accuracy and participants 1, 3, 4, 8, and 10 achieving the highest accuracy. The kappa values, which measure the agreement between observed and predicted classifications, range from 0.875 to 0.975, indicating excellent classification consistency. Precision and sensitivity metrics also demonstrate high performance, indicating the method’s effectiveness in identifying the correct class.

#### 3.1.2. MNE Method

Table 2 highlights the results for the same 10 participants using the MNE method. The accuracy spans from 83.33% to 98.33%, with participant 8 showing the lowest accuracy. The kappa values reflect strong classification agreement, with values ranging from 0.750 to 0.975. Notably, the precision and sensitivity are also consistently high, suggesting that MNE effectively captures relevant neural patterns for motor imagery tasks.

#### 3.1.3. Dipole Fitting Method

The results for the dipole fitting method, as shown in Table 3, reveal an accuracy range from 80.00% to 98.33%. Participant 9 had the lowest accuracy, indicating some variability in classification performance. The kappa values range from 0.700 to 0.975, reflecting good to excellent agreement. The precision and sensitivity metrics are robust, confirming the method’s capability to accurately classify different motor imagery tasks.

### 3.2. Overall Classification Performance across All Participants

Following the individual participant analysis, the average classification metrics across all 50 participants for each method are presented. Table 4 provides a detailed comparison, showcasing the significant improvements achieved by source localization methods compared to traditional sensor domain techniques.

The **dipole fitting** method achieved the highest overall classification accuracy at 91.03%, marking a substantial improvement over traditional methods like CSP + LDA, which achieved only 55.57%. This result demonstrates a 35.46% increase in classification accuracy, underscoring the precision of dipole fitting in capturing localized neural sources relevant to motor imagery tasks. Furthermore, the kappa score of 0.8655 for dipole fitting reflects a high level of agreement between predicted and actual classifications, with corresponding improvements in precision (0.9106) and sensitivity (0.9103), highlighting the method’s robustness.

**MNE**, with an accuracy of 89.07%, also outperformed traditional sensor domain methods significantly. Compared to CSP + LDA and FBCSP + SVM, which achieved accuracies of 55.57% and 57.57%, respectively, MNE shows an improvement of over 30%. Its kappa score of 0.8360 and strong precision (0.8916) and sensitivity (0.8907) further indicate that MNE provides a more comprehensive representation of the neural dynamics involved in motor imagery.

**Beamforming** achieved a slightly lower accuracy of 87.17% compared to dipole fitting and MNE, but still demonstrated a notable improvement over sensor domain techniques. The 87.17% accuracy represents a 31.60% improvement over CSP + LDA, with a kappa value of 0.8075. The beamforming precision (0.8734) and sensitivity (0.8717) results also highlight its effectiveness in identifying the active neural regions involved in motor tasks, though with a slightly broader focus compared to the more localized methods of dipole fitting and MNE.

In contrast, traditional methods, such as **CSP + LDA** and **FBCSP + SVM**, achieved lower classification accuracies of 55.57% and 57.57%, respectively, highlighting the limitations of sensor domain techniques in capturing the spatial complexity of neural activity. Even advanced sensor domain methods like **TSLDA + DGFMDRM** (61.20%) and **TWFB + DGFMDRM** (72.21%) failed to reach the classification performance of the source domain methods, indicating that transitioning to source-based analysis allows for a deeper and more precise understanding of the neural processes involved in motor imagery.

Overall, the results clearly demonstrate that source localization techniques—dipole fitting, MNE, and beamforming—achieve significantly higher classification accuracies, kappa scores, precision, and sensitivity than traditional sensor-based methods. These findings emphasize the effectiveness of transitioning to source-based analysis for improving the performance of EEG-based BCIs.

Figure 4 illustrates the confusion matrices for the dipole fitting, MNE, and beamforming methods. These matrices visually represent the classification performance, indicating the percentage of correct and incorrect predictions for each class (left hand, right hand, and rest). The high values along the diagonal of each matrix indicate strong agreement between the predicted and actual classes, further demonstrating the effectiveness of source localization techniques in motor imagery classification tasks.

In conclusion, the comparative analysis with traditional sensor domain methods, such as CSP + LDA and FBCSP + SVM, demonstrates the clear advantages of source domain techniques. While sensor domain methods typically yielded lower classification accuracies and performance metrics, source localization methods significantly improved the detection of motor imagery tasks by accurately capturing neural activity patterns. This enhancement in classification performance has important implications for developing more effective and reliable BCIs, particularly for clinical applications like stroke rehabilitation, where the precise detection of motor intentions is essential for successful therapeutic interventions.

## 4. Discussion

The results of this study underscore the superiority of source localization techniques, such as dipole fitting, MNE, and beamforming, over traditional sensor domain methods for classifying motor imagery (MI) tasks. Consistent with the findings of previous studies [13,21], our study shows that these advanced source-based methods significantly improve classification accuracy by providing a more accurate representation of neural activity at the cortical level. This shift from sensor-based to source-based analysis is crucial for enhancing the effectiveness of EEG-based BCIs, particularly in clinical applications like stroke rehabilitation.

### 4.1. Comparative Performance Analysis

Our comparative analysis between sensor domain methods (such as CSP + LDA and FBCSP + SVM) and source localization techniques reflects the broader consensus in the field [15]. As highlighted by previous research [10,20], sensor-based methods often yield lower classification accuracies due to their inability to capture the spatial complexities of brain activity. By contrast, source localization techniques, as evidenced in [16,23], offer greater spatial resolution, allowing for a more precise identification of neural sources related to MI tasks.

### 4.2. Detailed Results and Implications

The individual participant analysis further highlights the consistency and reliability of these methods. The variability in performance across participants was minimal, with most participants achieving high classification accuracies. This consistency suggests that source localization methods are not only more accurate but also more generalizable across different individuals, which is crucial for the practical application of BCIs.

The confusion matrices for the source localization methods showed high accuracy along the diagonal, indicating that these methods reliably distinguish between different MI tasks, including left-hand movement, right-hand movement, and rest states. This capability is essential for real-world BCI applications, where accurate task classification directly impacts the system’s effectiveness.

### 4.3. Applications in BCI and Neurorehabilitation

The improvements in classification accuracy achieved through source localization methods have substantial implications for the development of BCIs, especially in neurorehabilitation. The accurate classification of motor intentions is critical for facilitating motor recovery in stroke patients, as demonstrated in studies [11,12]. By providing real-time feedback on motor imagery, BCIs can encourage the use of affected limbs, thereby promoting functional recovery. Furthermore, studies [28,29] have shown that BCIs leveraging source localization techniques can engage neuroplasticity mechanisms, enhancing rehabilitation outcomes.

### 4.4. Strengths and Limitations

A key strength of this study is the application of source localization techniques, which resulted in significant improvements in classification accuracy. These results align with earlier work on the potential of such methods in EEG-based BCIs [17,19]. However, like many studies, this research has limitations. The relatively small dataset used and the lack of real-time implementation present challenges for translating these findings into clinical applications. Future work should focus on expanding the dataset and incorporating real-time training and validation to ensure broader applicability and clinical relevance.

### 4.5. Future Directions

Future research should aim to integrate these advanced source localization techniques into practical BCI systems, focusing on real-time implementation and user-friendly interfaces. Additionally, expanding the dataset to include a more diverse patient population will be essential for validating the generalizability of these findings across different neurological conditions. This expansion could also help refine the models further, improving their applicability in various clinical and assistive contexts.

## 5. Conclusions

In this study, we demonstrated that combining advanced source localization techniques (dipole fitting, MNE, and beamforming) with ResNet CNN architecture significantly improves the classification of motor imagery (MI) tasks in acute stroke patients, achieving classification accuracies of 91.03%, 89.07%, and 87.17%, respectively. These methods outperformed traditional sensor domain approaches like CSP + LDA, highlighting the critical advantage of transitioning from sensor to source domains for more precise cortical activity representation.

The main contributions of this paper include the following:**Enhanced classification accuracy**: source localization before CNN analysis improved classification accuracy, demonstrating more precise and reliable BCIs for clinical applications.**Advanced health technology integration**: the integration of advanced preprocessing, source localization, and deep learning supports personalized therapies for stroke patients by accurately interpreting brain signals.**Innovative use of source localization with CNN**: combining source localization techniques with ResNet enabled the effective capture of EEG spatial patterns, improving classification performance and advancing research in BCI and neurorehabilitation.

These contributions demonstrate the potential of our approach to advance BCI technology and improve neurorehabilitation outcomes for stroke patients.

## Figures and Tables

**Figure 1 bioengineering-11-00967-f001:**
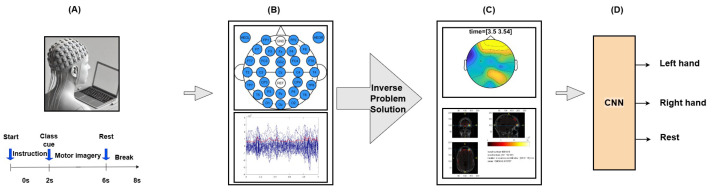
Flowchart of the proposed framework. (**A**) Participant setup: trial phases include preparation, motor imagery, and rest. (**B**) EEG data acquisition: 29 active electrodes and 2 EOG electrodes placed according to the 10-10 system. (**C**) Data analysis: Topoplots and source localization techniques (MNE, dipole fitting, beamforming) for cortical activity mapping. (**D**) Classification: ResNet-CNN classifies motor tasks using localized EEG data.

**Figure 2 bioengineering-11-00967-f002:**
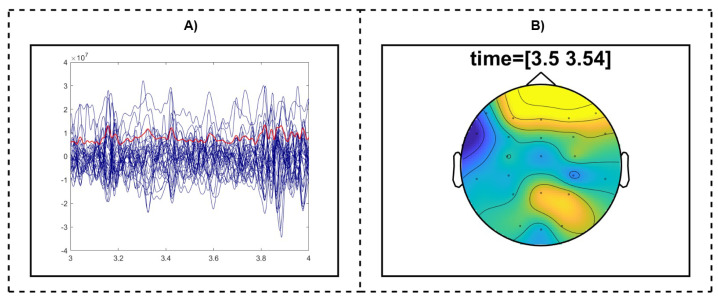
(**A**) EEG data showing the representation of a single trial (blue) and the global mean field power (red), illustrating the EEG signal dynamics and average neural activity. (**B**) Topographic map for the right-hand movement task, displaying the cortical distribution of neural activity and the time–frequency representations of beta (15–25 Hz) after the right-hand response.

**Figure 3 bioengineering-11-00967-f003:**
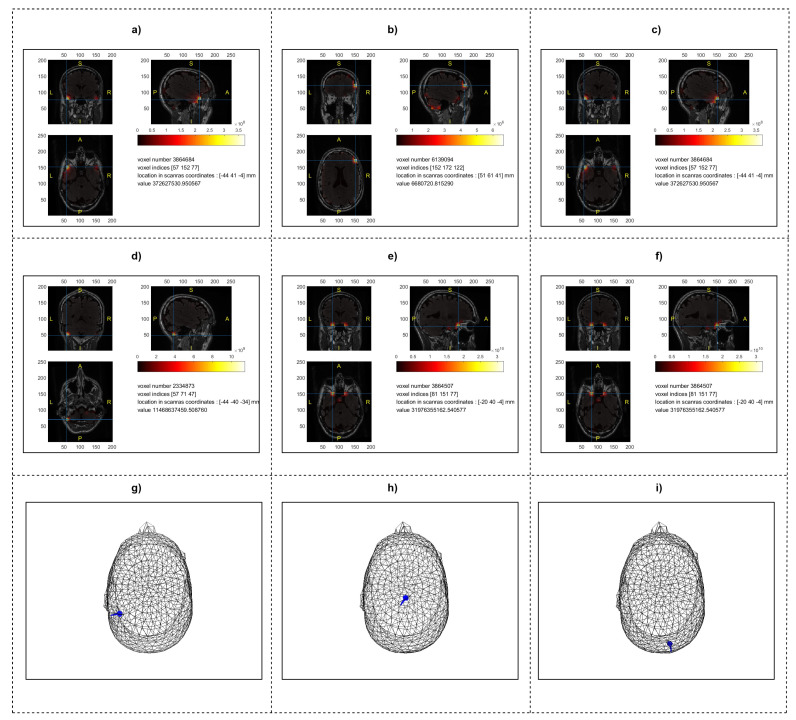
Source localization results for motor imagery tasks: (**a**–**c**) depict beamforming results, showing localized activity in the primary motor cortex (M1), supplementary motor area (SMA), and parietal cortex for the left-hand, right-hand, and rest conditions, respectively. (**d**–**f**) display MNE results, with a broader distribution of activity across the M1, premotor cortex, and parietal regions for the same tasks. (**g**–**i**) illustrate dipole fitting results, highlighting focal activation points in M1 and SMA, demonstrating the specificity of the neural sources involved in these tasks.

**Figure 4 bioengineering-11-00967-f004:**
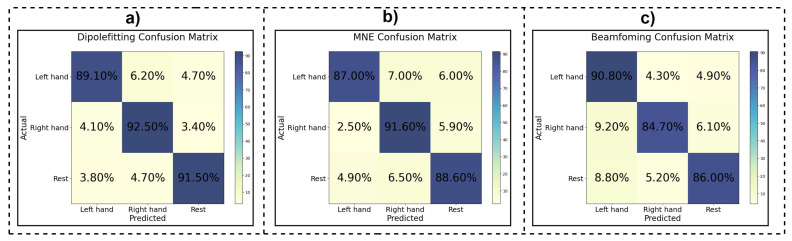
Confusion matrices for the (**a**) dipole fitting, (**b**) MNE, and (**c**) beamforming methods. The matrices show the percentage of correct and incorrect predictions for each class (left hand, right hand, and rest), highlighting the effectiveness of source localization techniques in improving classification accuracy for motor imagery tasks.

**Table 1 bioengineering-11-00967-t001:** Classification performance for the beamforming method (top 10 participants). The table shows the accuracy, kappa, precision, and sensitivity for each participant.

Participant	Accuracy (%)	Kappa	Precision	Sensitivity
participant 1	98.33	0.975	0.984	0.983
participant 2	96.67	0.950	0.967	0.967
participant 3	98.33	0.975	0.984	0.983
participant 4	98.33	0.975	0.984	0.983
participant 5	96.67	0.950	0.970	0.967
participant 6	91.67	0.875	0.920	0.917
participant 7	96.67	0.950	0.968	0.967
participant 8	98.33	0.975	0.984	0.983
participant 9	96.67	0.950	0.970	0.967
participant 10	98.33	0.975	0.984	0.983

**Table 2 bioengineering-11-00967-t002:** Classification performance for the MNE method (top 10 participants). The table shows the accuracy, kappa, precision, and sensitivity for each participant.

Participant	Accuracy (%)	Kappa	Precision	Sensitivity
participant 1	91.67	0.875	0.926	0.917
participant 2	93.33	0.900	0.944	0.933
participant 3	96.67	0.950	0.970	0.967
participant 4	98.33	0.975	0.984	0.983
participant 5	98.33	0.975	0.984	0.983
participant 6	88.33	0.825	0.914	0.883
participant 7	95.00	0.925	0.957	0.950
participant 8	83.33	0.750	0.873	0.833
participant 9	88.33	0.825	0.914	0.883
participant 10	93.33	0.900	0.944	0.933

**Table 3 bioengineering-11-00967-t003:** Classification performance for the dipole fitting method (top 10 participants). The table shows the accuracy, kappa, precision, and sensitivity for each participant.

Participant	Accuracy (%)	Kappa	Precision	Sensitivity
participant 1	88.33	0.825	0.888	0.883
participant 2	93.33	0.900	0.944	0.933
participant 3	83.33	0.750	0.853	0.833
participant 4	91.67	0.875	0.924	0.917
participant 5	95.00	0.925	0.951	0.950
participant 6	93.33	0.900	0.935	0.933
participant 7	88.33	0.825	0.892	0.883
participant 8	93.33	0.900	0.939	0.933
participant 9	80.00	0.700	0.829	0.800
participant 10	98.33	0.975	0.984	0.983

**Table 4 bioengineering-11-00967-t004:** Comparison of average classification performance of various methods. The first four methods, CSP + LDA, FBCSP + SVM, TSLDA + DGFMDRM, and TWFB + DGFMDRM, represent state-of-the-art techniques for EEG-based motor imagery classification, as cited in [14]. The last three rows present our proposed source localization methods, dipole fitting, MNE, and beamforming, demonstrating significant improvements in classification accuracy, kappa, precision, and sensitivity.

Method	Average Accuracy (%)	Kappa	Precision	Sensitivity
CSP + LDA [15]	55.57	0.1114	0.5619	0.5707
FBCSP + SVM [15]	57.57	0.1514	0.5690	0.5668
TSLDA + DGFMDRM [15]	61.20	0.2240	0.6160	0.6111
TWFB + DGFMDRM [15]	72.21	0.4442	0.7543	0.7845
**Dipole fitting**	**91.03**	**0.8655**	**0.9106**	**0.9103**
**MNE**	**89.07**	**0.8360**	**0.8916**	**0.8907**
**Beamforming**	**87.17**	**0.8075**	**0.8734**	**0.8717**

## Data Availability

The data used in this paper are publicly available at https://doi.org/10.6084/m9.figshare.21679035.v5. The dataset is a publicly available dataset [15].

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
