# Peer review of "Integrating Electroencephalography Source Localization and Residual Convolutional Neural Network for Advanced Stroke Rehabilitation"

_bioengineering, 2024, doi:10.3390/bioengineering11100967_

Round 1

Reviewer 1 Report

Comments and Suggestions for Authors

In this manuscript, the authors introduced a customized ResNet CNN incorporated with source localization techniques, including MNE, dipole fitting, and beamforming to identify cortical activity. Their method achieved great accuracy outperforming traditional methods. The overall quality of this manuscript is good. 

I have a suggestion to make this paper more complete: it would be great if the authors could cite some basic papers for the key methods including ResNet, source localization techniques, etc. For example, Kaiming He, Xiangyu Zhang, Shaoqing Ren, Jian Sun, "Deep Residual Learning for Image Recognition" for ResNet near line 196. 

Author Response

Please see the attached responses.

Reviewer 2 Report

Comments and Suggestions for Authors

The work entitled "Integrating EEG Source Localization and ResNet CNN for Advanced Stroke Rehabilitation" by Kaviri and Vinjamuri is a very informative work that studies the use of source localization EEG methods to increase precision, accuracy, etc. to classify motor imagery tasks, such as left, right hand movements and at rest. The method seems to be more effective when compared with  other traditional EEG methods based on sensor domains. I recomend its publication after the authors answer the following points:

1) In the introduction, please define all the acronyms. For example SVM, not all readers are experts in the filed.

2) It is not clear in section 2.1 and later sections, why the authors choose the beta band (15-25 Hz) in many figures. What happens on other bands?

3) Figure 2 caption. is it right-hand response or finger?

4) What it the size of the whole data base used in this work? How many signals were used for learning? How many for assesing the effectiveness of the method? Please clarify.

5) Why did you use a data base with stroke patients? it would not be enough using normal people to show the article's point? It is not clear why, please clarify.

6) What would be the outcome if you were using non-stroke subjects?

Author Response

Please see the attached responses.

Reviewer 3 Report

Comments and Suggestions for Authors

Kaviri and collaborators looked at an EEG source-localization approach to classifying motor tasks in people who have suffered a stroke. Their results seems to demonstrate that their approach is accurate and effective. I can see the clinical importance of this study, but the manuscript needs a lot of work and I hope my comments below will help improve it.

Abstract: will need to be updated base don my comments below.

Introduction

  • "The high incidence of stroke" Can you please remind the reader what this incidence is in the general population?
  • "EEG signals are crucial for BCIs, providing communication methods for individuals with severe neuromuscular disorders." This statement is not clear and seems unrelated to the paragraph. Actually, the last sentence of this paragraph is sufficient to convey the message. Please consider deleting the first two sentence.
  • Following from my previous comment, the introduction is very long and we get lost in details that are not needed for the understanding of the aims of the current study. The introduction needs to be shortened to present only the background needed to understand the aims.
  • Please define SVM, as well as 'synchronous' BCI. Also define CNN and MNE, as well as dipole fitting and beamforming
  • I'm surprised to see results in the introduction. Also, the main contribution of the paper belongs to the conclusion. The structure of the paper is also not needed here. What the reader needs to see is a clear aim and hypothesis at the end of the introduction. 

Methods

  • The structure of this section is very confused. Please present the design of the study and participants first. This first section should include details about study setting, recruitment, consent and ethics approval. Then present the technical information about the system and finally present the analyses methods. A good structure will help limiting repetitions as there is a lot of it at the moment. 
  • Information about the system should be in the text, and does not need to be repeated in Figure 1 legend. Similarly, what is depicted on the figure should be explained in the legend, not in the text. I'm also not convinced that this figure is needed.
  • Please explain 'class cue'

Results

  • Information about the system and about analyses should be in the methods section.
  • Results should begin with a description of the sample. Age, type of stroke, side of the body affected (this is extremely important for this study and should be taken into consideration in the analyses)
  • Numerical results: why only a subset of 10 participants?

General comment:

  • Please refrain from using the term 'subject' which refers to the person begin subjected to something. 'Participant' is preferred as it denotes the active and willing contribution of the person.
  • So many techniques have been used, lots of acronyms, I'm lost, I don't know what I'm meant to look at. The methods section did not prepare me for so many different results.

Discussion:

  • "The results of this study clearly demonstrate [...]": I must admit that this is not clear to me as the results section is presented currently.
  • "These findings underscore the value of transitioning from sensor-based to source-based analyses in EEG data interpretation, particularly in BCI applications." Was it an aim to demonstrate this? I don't remember the introduction mentioning anything about sensor-based analyses.
  • The discussion seems to highlight the main results, but these results should be discussed against the literature.
  • A 'strength and limitations' section is missing.
  • The conclusion is too long, please only highlight the essential.

Author Response

Please see the attached responses.

Round 2

Reviewer 3 Report

Comments and Suggestions for Authors

Thanks for considering my comments. However, not all of them have been addressed to satisfaction.

  • I still think the introduction too long and goes into unnecessary details, but I'll leave it with the Editor to decide.
  • Please delete the paragraph starting with "The rest of the paper [...]", this is not needed.
  • Methods: there are still acronyms that have not been described. Deciding whether acronyms are described or not is not about the reader being expert in the field or not as suggested by the authors, it is about the fact that acronyms should always be described at first presentation in the text.
  • There's still a lot of repetition in the methods, a few examples are: repeating the number of participants, repeating that the EEG follows 10-10 systems, and more. Please read again carefully and avoid repetitions.
  • "The technical details related to the system setup and analysis are integral to understanding how the results were obtained. While these are typically found in the methods section, we included them here to ensure a direct connection between the methodology and the results being discussed. This structure allows for a clearer understanding of how the system influences the findings. However, we acknowledge your point and will consider relocating this information to the methods section in future revisions if necessary."

Yes please, consider relocating the first paragraph, as well as section 3.1. The reader would be looking for this information in the methods section.

  • I still think that the results section is confusing and difficult to read and again, I will let the final decision with the Editor.
  • "We will revise the discussion to include a comparison of our results with relevant literature, providing context and highlighting how our findings align with or differ from previous studies."

I do not see these improvements. 

Author Response

Please see the attachment below.
